# Hotoda’s Sequence and Anti-HIV Activity: Where Are We Now?

**DOI:** 10.3390/molecules24071417

**Published:** 2019-04-10

**Authors:** Valeria Romanucci, Armando Zarrelli, Giovanni Di Fabio

**Affiliations:** Department of Chemical Sciences, University of Napoli Federico II, Via Cintia 4, I-80126 Napoli, Italy; zarrelli@unina.it

**Keywords:** anti-HIV agent, G-quadruplex, aptamer, Hotoda’s sequence, modified sequences

## Abstract

The pharmacological relevance of ODNs forming G-quadruplexes as anti-HIV agents has been extensively reported in the literature over the last few years. Recent detailed studies have elucidated the peculiar arrangement adopted by many G-quadruplex-based aptamers and provided insight into their mechanism of action. In this review, we have reported the history of a strong anti-HIV agent: the 6-mer d(TGGGAG) sequence, commonly called “Hotoda’s sequence”. In particular, all findings reported on this sequence and its modified sequences have been discussed considering the following research phases: (i) discovery of the first 5′-modified active d(TGGGAG) sequences; (ii) synthesis of a variety of end-modified d(TGGGAG) sequences; (iii) biophysical and NMR investigations of natural and modified Hotoda’s sequences; (iv); kinetic studies on the most active 5′-modified d(TGGGAG) sequences; and (v) extensive anti-HIV screening of G-quadruplexes formed by d(TGGGAG) sequences. This review aims to clarify all results obtained over the years on Hotoda’s sequence, revealing its potentiality as a strong anti-HIV agent (EC_50_ = 14 nM).

## 1. Introduction

Guanine quadruplexes (G4s) are a class of non-canonical nucleic acid structures that have attracted considerable attention thanks to their structural stability under physiological conditions. This class is characterized by an extensive conformational polymorphism that amplifies its biological potentialities. Different G4 topologies have been identified by variations in strand stoichiometry and polarity, as well as by the nature and length of loops and their location in the sequence [1,2].

The increasing interest in the G4 role in gene regulation and its active role in telomerase maintenance has highlighted the biological relevance of G4 structures [3,4,5]. A large variety of potential G4s have been identified in the human genome that could perform important biological roles [6,7,8]. In addition to humans, G-quadruplexes have been found in other mammalian genomes [9], yeasts [10], protozoa [11], bacteria [12,13] and viruses [14].

During the last few years, G4s able to recognize and bind specific molecules acting as aptamers have been developed. In this regard, the peculiar arrangements of G-quadruplexes seem to be a crucial requirement for aptamer recognition, thanks to the elevated specificity and selectivity of G4 structures. G-quadruplexes have been exploited for therapeutic strategies as anti-HIV agents, and they have been successfully designed to target the HIV virus at different stages of its life cycle. The potential targets of HIV infection are HIV-1 reverse transcriptase, HIV RNase H, HIV-1 integrase (IN) and the viral surface glycoprotein known as gp120 [14,15]. Among the first-discovered G-quadruplex aptamers targeting the gp120 glycoprotein as a primary target and the HIV-1 integrase was the parallel-stranded G-quadruplex T30177 (Zintevir^TM^ developed by Aronex Pharmaceuticals in 1996), which showed high inhibition at nanomolar levels (IC_50_ = 100 nM) [16]. Subsequently, similar to Zintevir, other G-quadruplex aptamers, such as T30923 [17] and 93del [18,19] and the phosphorothioate ISIS-5320 [20], have been proven effective at inhibiting the HIV-1 virus. Interestingly, many researchers have discovered that anti-HIV G-quadruplexes adopt a parallel G4 arrangement.

In this context, in the 1990s, a group of Japanese researchers encouraged by recent discoveries concerning anti-HIV aptamers forming parallel G-quadruplexes reported an anti-HIV study on ODNs in which the 5′-end was covalently linked to a 4,4′-dimethoxytriphenylmethyl (DMT) residue. This discovery was facilitated by easy access to DMT-modified ODNs that are generally obtained as intermediate products of automated DNA synthesizers [21,22].

This study represents the “mother-generating” of the history of Hotoda’s sequence, because, for the first time, the active sequence d(^5′DMT^TGGGAGGTGGGTCTG^3′^) (SA-1042) was discovered, along with the importance of the 5′ end modification for the expression of anti-HIV activity. Subsequent studies highlighted that the active sequence was the 5′-end tract of SA-1042, d(^5′DMT^TGGGAG^3′^), identified as SA-1080 [23]. Since then, many modifications have been proposed for d(TGGGAG), the lead sequence commonly named “Hotoda’s sequence” (Figure 1a).

In this review, a critical overview of all data reported from 1994 through to today about anti-HIV aptamers based on Hotoda’s sequence has been presented. The goal is to clarify the relationship between the anti-HIV activity and the nature of the modified sequences studied during these years, as well as the evolution of associated knowledge from the 1990s, when these special G4 structures were discovered, until today. This paper also explains that through the comparison of thermodynamic and kinetic studies with biological assays, it was possible to understand the G4 behavior of modified sequences, and only in recent studies has the strong anti-HIV activity of Hotoda’s sequence d(TGGGAG) been discovered (EC_50_ = 14 nM).

## 2. The Discovery of the First 5′-End Modified of Hotoda’s Sequence: d[DMT-(TGGGAG)] and R-95288

In the 1990s, the active sequence d(TGGGAG) conjugated with a DMT group at the 5′-end was identified as being able to inhibit the HIV-1 virus in the micromolar range (Figure 1b, SA-1080, IC_50_ = 1.0 μM) [23]. Additionally, the mechanism of action against the virus was reported. In fact, the modified hexamer was able to inhibit the binding of the glycoprotein gp120 to its receptor CD4 molecule, blocking the early stage of the HIV-1 infection process.

The 6-mer d(TGGGAG) was defined as an aptamer able to recognize and bind both the V3 loop and the CD4-binding site on viral gp120. In this paper, the authors described the structural requirement of the modified ODN for exhibiting anti-HIV activity that consists of the DMT group conjugation at the 5′-end of the sequence and the importance of at least three consecutive guanine residues, which are essential for the formation of a guanine cluster. At that time, the formation of a G4 structure by the DMT conjugated d(TGGGAG) sequence had not yet been identified. In this context, also very recently, Zeye H. and co-workers [24] reported a deepened study on the aptameric recognition of some 3′-end modified G4s based on the d(^5′DMT^TGGGAG^3′^) sequence with HIV-1 glycoproteins gp120 and gp41. The authors demonstrated, using Surface Plasmon Resonance analyses, an enhanced interaction of these inhibitors with respect to the lead sequence used [d(^5′DMT^TGGGAG^3′^)]. 

In the same years, Hotoda and co-workers reported the synthesis of a mini-library based on the 5′-phosphodiester of d(TGGGAG) conjugated with different aromatic residues. From the anti-HIV screening of different modified 6-mer ODNs, no more active modified sequences were discovered [25]. Meanwhile, the same researchers reported the synthesis and anti-HIV-1 activity of a mini library of d(TGGGAG) with a 5′,3′-bis-conjugated lead sequence, wherein the 5′-end conjugation was carried out by an ether bond and the 3′-end by a phosphodiester bridge [26]. Among the various 3′- and 5′-end-modified 6-mers tested, the researchers identified a potent inhibitor, R-95288, with high anti-HIV-1 activity (IC_50_ = 1.0 μM) and low toxicity. The 6-mer (R-95288) bearing a 3,4-DBB group at the 5′-end and a 2-hydroxyethylphosphate group at the 3′-end was also the most stable when incubated with mouse or human plasma.

To explore the anti-HIV activity and stability in the human plasma of R-95288, some modifications at phosphate groups (phosphorothioate, phosphoramidate and methylphosphonate bonds) were proposed by Koizumi et al. [27]. The findings confirmed major activity of the natural phosphate R-95288, even though major stability was found for some phosphorothioate derivatives. 

Only with a SAR study conducted on a similar family of oligonucleotides was the most potent anti-HIV agent finally discovered: a performing version of R-95288 containing only the modification of a 3,4-DBB group at the 5′-end (Figure 1b, R-95288, IC_50_ = 0.37 μM) [28].

In addition, in this study, it was determined that the anti-HIV-1 activity of modified sequences first requires conjugation with an aromatic substituent of adequate size at the 5′-end and the formation of stable G-quadruplex structures that, in agreement with findings for ISIS 5320, could play an important role in aptamer recognition [20,29].

In this context, with the discovery of the R-95288 aptamer, the CD profiles of modified sequences were characterized, together with melting temperature analyses. The parallel topology of the G4 structures based on the R-95288 sequence and other modified d(TGGGAG) sequences was attributed on the basis of the characteristic positive maximum at 260 nm and the negative minimum close to 240 in the relative CD spectrum.

Subsequently, some modification of guanines within this sequence have been reported. In particular, Koizumi et al. described the N^2^-methylation of guanine, which mainly led to enhancements of the G-quadruplex thermal stability [30]. In contrast, the modification proposed by Jaksâ consisting of the replacement of dG residues with non-natural 8-aza-3-deaza-2′-deoxyguanosine monomers led to a significant decrease in antiviral activity [31].

### Biophysical and NMR Studies on Hotoda’s Sequence and Its Modified Sequences

An interesting study that aimed to explore possible relationships between the specific effects induced by terminal groups and both the thermodynamic and kinetic aspects of G-quadruplex formation was reported by D’Onofrio et al. [32].

Considering the inactivity of Hotoda’s d(TGGGAG) sequence and the low antiviral activity found for 5′-modified d(TGGGAG) sequences with small aromatic residues [28], the authors supposed that the antiviral activity was therefore attributed to the presence of large aromatic groups at the 5′-end, which are able to better stabilized G-quadruplex structures by hydrophobic interactions. DSC and CD characterization carried out on a set of active 5′-modified d(TGGGAG) sequences suggested a correlation between the G4-stability, rate of G4 formation and anti-HIV activity of 5′-modified ODNs. In particular, they examined conjugation at the 5′-end with DMT, tert-butyldiphenylsilyl (TBDPS) and 3,4-dibenzyloxybenzyl (DBB, R-95288 sequence) groups. Biophysical characterization disclosed that even if the natural sequence forms a G-quadruplex structure, the structure presents a lower thermal stability (T_m_ = 41 °C) with respect to G4s based on 5′-modified sequences. The authors concluded that large aromatic groups play a crucial role in the enhancement of thermal stability, supposing that the kinetics of G-quadruplex folding can also be influenced by these groups. These two factors are reputed to be responsible for the strong anti-HIV activity.

An NMR study was also carried out on Hotoda’s sequences in 2012 by Virgilio and co-workers, who observed the coexistence of multiple species supposed to be higher-order aggregates that could be favored by the formation of a 3′-end G-tetrad [33]. The authors solved the NMR experimental problem by the introduction of a thymine residue at the 3′-end, which avoids the formation of aggregates and permits the characterization of the unique G4 adopted by the unmodified Hotoda’s sequence.

Unfortunately, they did not characterize the active aptamer with the DMT group at the 5′-end because of the lipophilicity of this aromatic group, which also prevented the formation of an investigable unique G-quadruplex structure, even with the addition of a 3′-end thymine residue. Finally, in this NMR study, the researchers mainly described, for the first time, the formation of an A-tetrad between the G-tetrads of Hotoda’s d(TGGGAG) sequence.

## 3. Modified d(TGGGAG) Hotoda’s Sequences

### 3.1. End-Phosphodiester ODN Conjugates Based on Hotoda’s Sequence

In these years, D’onofrio et al., investigated the end-conjugation with sugars for the d(TGGGAG) sequence as a strategy to improve the pharmacological profile of these oligonucleotides [34]. On the basis of thermal denaturation studies on the resulting G-quadruplexes, they reported that especially the insertion of a mannose residue at the 3′-phosphate end improves the stability of the G-quadruplex complex and confers anti-HIV activity to the unmodified sequence. In contrast, 5′-tethering with the same monosaccharides in all cases decreases both the G-quadruplex stability and the antiviral activity. These studies underlined the crucial role of aromatic groups at the 5′-end. 

In this context, since 2011, aiming to expand the repertoire of potential end-modified d(TGGGAG) aptamers, Di Fabio and co-workers reported a series of modified d(TGGGAG) conjugated with aromatic and non-aromatic groups at the 5′-end by a phosphodiester bond (Figure 2) [35,36]. 

The main goal was understanding the role of the 5′-end group in G4 folding and in the anti-HIV activity. They reported that both aromatic and non-aromatic 5′-end-modified sequences formed parallel tetramolecular G-quadruplexes with increased thermal stability (Tm = 57–90 °C), but not all sequences showed high anti-HIV activity (Figure 2).

These results called into question what had previously been reported by D’Onofrio (2007) regarding the strong dependence between the G4 thermal stability and the anti-HIV activity [32]. In fact, this paper claimed that the major stability of the investigated G4 complexes was not directly correlated with their pronounced anti-HIV activity.

However, more importantly, the authors discovered a more active modified Hotoda’s sequence conjugated with a p-benzyloxyphenyl group (PBP) at the 5′-end and endowed with anti-HIV activity in the nanomolar range (Figure 2, EC_50_ = 61 nM). Subsequently, following the same synthetic approach of Di Fabio et al., a 5′-end cholesteryl-HEG (hexaethylene glycol) derivative of Hotoda’s sequence was reported, but the anti-HIV activity of the sequence was not evaluated [37].

### 3.2. Modifying the Nucleobases of the d(TGGGAG) Sequence

Following the results reported by Di Fabio and co-workers, several publications reported the synthesis and biophysical and biological characterization of new modified Hotoda’s sequences created by remodeling the nucleobases of the sequence. Pedersen et al. described how the incorporation of LNA (locked nucleic acids) or INA/TINA monomers (intercalating nucleic acids and twisted intercalating nucleic acids) that present pyrene bearing moieties (Figure 3a) into terminal and internal positions of the sequence influences both the G-quadruplex arrangement and the anti-HIV activity of the 5′-end DMT Hotoda’s sequence [38].

The study data confirmed the relevancy of the 5′-end DMT group in the anti-HIV activity and pointed out that the incorporation of an LNA monomer enhances both the thermal stability of G-quadruplex structures and the anti-HIV activity. Moreover, the insertion of a pyrene-bearing INA moiety alters the G4 arrangement, resulting in inactive compounds, whereas the TINA monomer was found to enhance the activity.

In the same year, Chen et al. also reported a brief study on Hotoda’s sequences modified at the 5′-end thymidine nucleobase with TBDPS groups. The study reported high anti-HIV activities of the new modified sequences, identifying a new position for the conjugation of hydrophobic groups (Figure 3b, **I**, IC_50_ = 0.33 µM) [39].

An extension of this paper was recently reported by the same research group [40]. The authors elucidated the synthesis and introduction into Hotoda’s sequence of new nucleoside analogues by modifying the base with a hydrophobic *p*-benzyloxyphenyl group. All nucleobase-modified d(TGGGAG) sequences showed moderate anti-HIV activities (Figure 3b, **II**, IC_50_ = 0.78 µM).

### 3.3. Monomolecular and Bimolecular G-Quadruplexes Formed by Modified Hotoda’s Sequences

Aiming to improve the kinetics of G4 folding, Oliviero et al. reported the synthesis of tetra-end-linked oligonucleotides (TEL-ODNs) forming monomolecular G4s based on Hotoda’s sequence [41]. These new structures presented a special linker at the 3′-end that promotes the formation of a parallel monomolecular G4 structure with respect to the tetramolecular G4 formed by Hotoda’s sequences. This modification mainly provided significant improvement of the G4-folding kinetics, which were instead very low for tetramolecular structures.

In addition, the authors confirmed an enhancement of the thermal stability when the 5′-end of TEL-ODNs is conjugated with a lipophilic group (TBDPS). The anti-HIV data of 5′-end-conjugated TEL-ODNs based on Hotoda’s sequence (Figure 4a, **I**, EC_50_ = 82 nM) disclosed the importance of kinetic factors, together with conjugation at the 5′-end, in relation to the anti-HIV activity.

An extension of the TEL-ODNs repertoire was reported in 2012 by the same authors, in which they synthesized new TEL-ODNs to study the effect of the base sequence on the anti-HIV activity [42]. All new TEL-(TGGGXG)_4_ aptamers retained potent anti-HIV-1 activity in the nanomolar range. In particular, the replacement of adenosine by cytosine allows improvement of the anti-HIV activity (Figure 4a, **II**, EC_50_ = 39 nM).

Di Fabio et al. also reported the synthesis and biophysical characterization of a new family of modified d(TGGGAG) sequences, the so-called hairpins, that consists of bimolecular d(TGGGAG) G-quadruplexes conjugated at the 5′-end with aromatic groups (Figure 4b) [43].

The goal was the discovery of more active G4s based on Hotoda’s sequence by modulating the kinetic and thermodynamic factors of G4 folding. The hairpins are characterized by a hexaethylene glycol (HEG) linker at the 3′-end and different aromatic moieties at the 5′-end. Strong thermal stability was found for all hairpins. Moreover, conjugation with a lipophilic group conferred anti-HIV activity to the sequence (Figure 4b, **H_2–4_**). These results confirmed the relevance of 5′-end conjugation in the anti-HIV activity.

### 3.4. Kinetics Study and Anti-HIV Screening of the Most Active Hotoda’s Modified Sequences

To clarify what makes these oligonucleotides strong anti-HIV agents and whether the kinetics of G-quadruplex folding are a critical aspect in the expression of HIV-activity, Romanucci et al. reported a comprehensive kinetics study on the G4 folding of the most active 5′-end-modified d(TGGGAG) sequences using electrospray mass spectrometry (ESI-MS) [44].

From this study, they found very slow kinetics for the folding of tetramolecular G-quadruplexes, together with a strong effect of ODN concentrations (single-stranded concentrations) on the G4-folding kinetics. In addition, they clarified that both inactive natural Hotoda’s sequences and the most active 5′-end-modified d(TGGGAG) sequences possess the same kinetics, highlighting that the presence of a 5′-end group is not implicated in the kinetics of G4 folding.

Finally, ESI-MS experiments surprisingly revealed the formation of higher-order G4 structures unambiguously identified as octameric complexes.

In light of these findings concerning the modified Hotoda’s sequence, the same authors designed an extensive anti-HIV screening of the kinetically studied 5′-end-modified d(TGGGAG) sequences in comparison with the unmodified Hotoda’s sequence [45]. The experiments were performed by testing all ODNs at different single-stranded ODN concentrations only after 14 days from the start of the G4 folding process to evaluate the effects of both the time and the ODNs concentration on the amount of active G-quadruplex specie on the expression of anti-HIV activity. 

The anti-HIV activity and cytotoxicity of the ODNs were evaluated against wild-type (WT) HIV-1 strain IIIB and NL4.3, the T30177 (Zintevir)-resistant strain derived from NL4.3 (NL4.3/T30177), and the dextran sulphate 8000 (DS8000)-resistant strain derived from NL4.3 (NL4.3/DS8000), in MT-4 cell cultures.

In the latter case, the authors aimed to investigate the mechanism of action of these ODNs, which could be very similar to that reported by Este et al. for Zintevir, thanks to their structural similitude [16]. From this anti-HIV screening, the authors reported a similar mode of action of Zintevir for some 5′-end-modified d(TGGGAG) ODNs, suggesting different mechanisms depending on the modification at the 5′-end of d(TGGGAG) sequences.

Interestingly, they also tested the unmodified Hotoda’s sequence at different initial ODN concentrations against WT viruses and Zintevir- and DS8000-resistant strains. In this regard, a significant result reported in this paper was the strong anti-HIV activity of the unmodified Hotoda’s sequence, which, contrary to extensive reporting over the years by many publications, showed the highest anti-HIV activity (EC_50_ = 14 nM).

In detail, the Hotoda’s sequence without any modification and prepared under proper experimental conditions that favor G4 formation, displayed the highest anti-HIV activity. The formation of a parallel G-quadruplex structure was confirmed by CD and native gel electrophoresis analyses. 

Finally, the Hotoda’s sequence seems to follow a different mechanism of action with respect to its associated 5′-end-modified sequences.

## 4. Conclusions

In this review, the history, since its discovery in the 1990s, of a potent anti-HIV agent, the so-called Hotoda’s sequence, has been examined in depth. This sequence was reported to inhibit the primary target of HIV infection, binding to the envelope glycoprotein gp120 by aptameric recognition only when conjugated at the 5′-end with a lipophilic group (Figure 1b, R-95288; IC_50_ = 0.37 μM). Its significant anti-HIV activity was ascribed to two main factors: the peculiar arrangement in a parallel tetramolecular G-quadruplex structure and conjugation with a lipophilic group at the 5′-end of the d(TGGGAG) sequence.

Since this discovery, many groups have focused their research on the synthesis of new and effective modified Hotoda sequences, aiming at understanding the role of both the lipophilic 5′-end group and the G-quadruplex arrangement with respect to the strong anti-HIV activity observed. A biophysical characterization was performed on the first 5′-end-modified sequences, highlighting that the thermal stability of the resulting G-quadruplexes was higher than that reported for the inactive natural Hotoda’s sequence. This study suggested the implication of thermal stability in the anti-HIV activity of modified d(TGGGAG) sequences, but the theory was subsequently confuted by other investigations of different modified sequences.

Over the years, various very potent modified Hotoda’s aptamers have been found, and the most active sequences are a tetramolecular G4 based on Hotoda’s sequence and conjugated at the 5′-end with a *p*-benzyloxyphenyl group (Figure 2, EC_50_ = 61 nM) and the monomolecular G4 based on the d(TGGGCG) sequence conjugated at the 5′-end with a TBDPS group (Figure 4a, EC_50_ = 39 nM).

In the meantime, various modifications were reported, ranging from the introduction of 5′- and 3′-end linkers to transform the tetramolecular G-quadruplex arrangement into more kinetically favourable bimolecular and monomolecular arrangement to modification of the phosphate backbone and nucleobases. Additionally, the G4 folding kinetics of some active 5′-end-modified d(TGGGAG) sequences were investigated, highlighting no differences arising from conjugation with a lipophilic group at the 5′-end.

All these results have led researchers to conduct deepened screening of the anti-HIV activity of both natural and 5′-end-modified Hotoda’s sequences. Very interestingly, this study revealed the great potency of Hotoda’s sequence, in disagreement with the conclusion claimed before. The discovery of the high activity of the unmodified Hotoda’s sequence (EC_50_ = 14 nM), makes this sequence one of the most potent G-quadruplex anti-HIV agents today, highlighting how the study of these supramolecular structures is complex and needs increased attention from researchers. The promise of Hotoda’s sequence as an anti-HIV agent in upcoming clinical trials is alluring, and the challenges are considerable, but not insuperable.

## Figures and Tables

**Figure 1 molecules-24-01417-f001:**
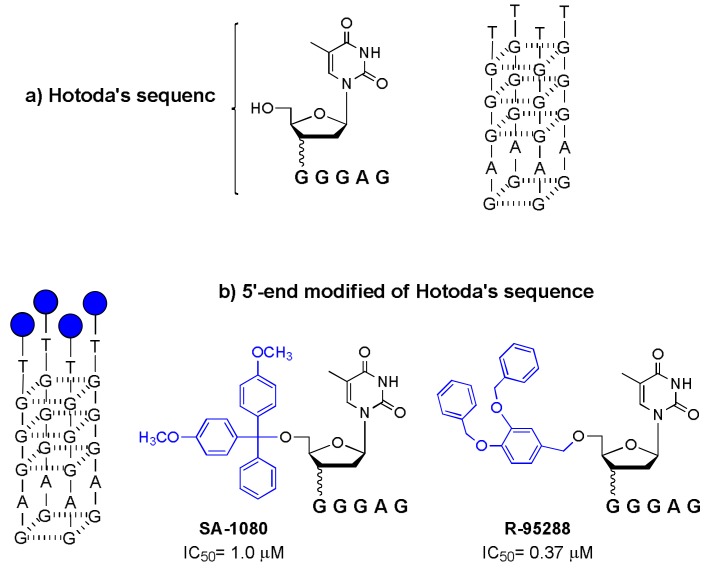
Schematic representation of: (**a**) Hotoda’s sequence; (**b**) its first 5′-end modified sequences with strong anti-HIV activity (IC_50_) [23,28].

**Figure 2 molecules-24-01417-f002:**
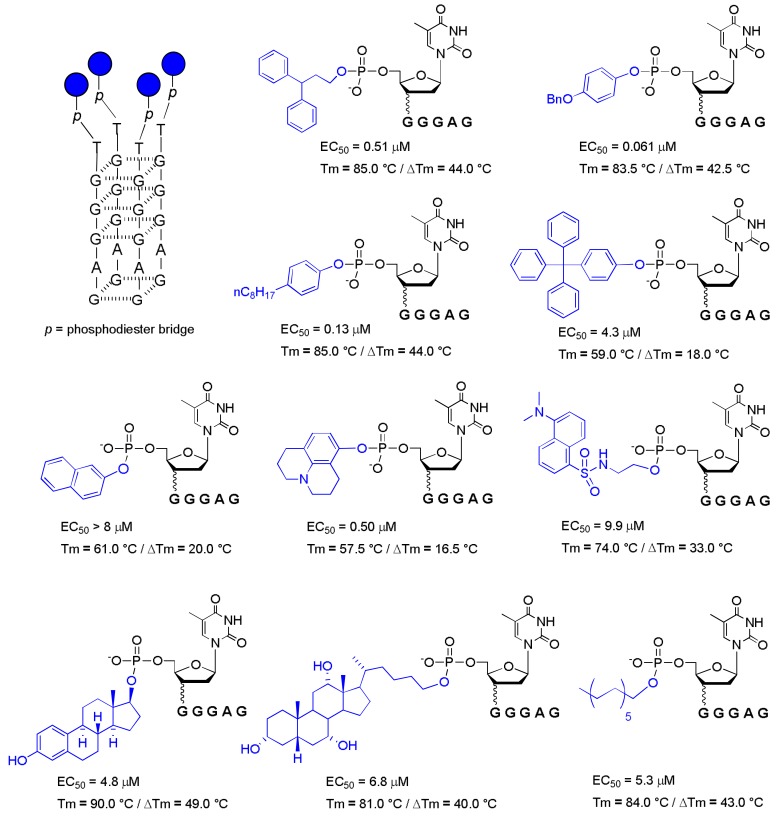
Anti-HIV activity (EC_50_), thermal stability (T_m_) and ∆T_m_ (the difference between the T_m_ of modified sequence and that of the natural one) of some 5′-end modified d(TGGGAG) sequences [35,36].

**Figure 3 molecules-24-01417-f003:**
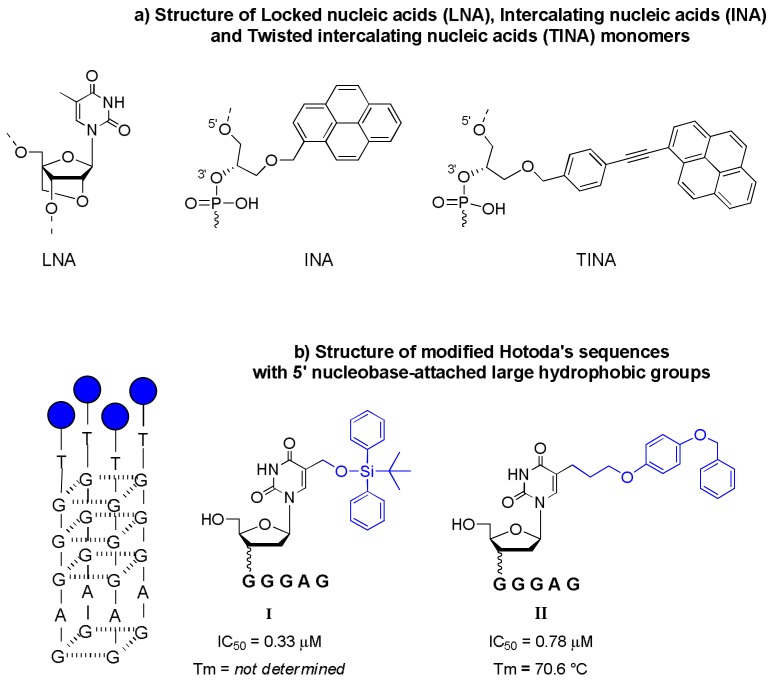
Schematic representation of the modified Hotoda’s sequences at nucleobases: (**a**) insertion of LNA, INA and TINA monomers in different position of d(TGGGAG) sequence [38]; (**b**) the most active modified d(TGGGAG) sequences with the insertion of aromatic groups at 5′-end thymidine nucleobase [39,40].

**Figure 4 molecules-24-01417-f004:**
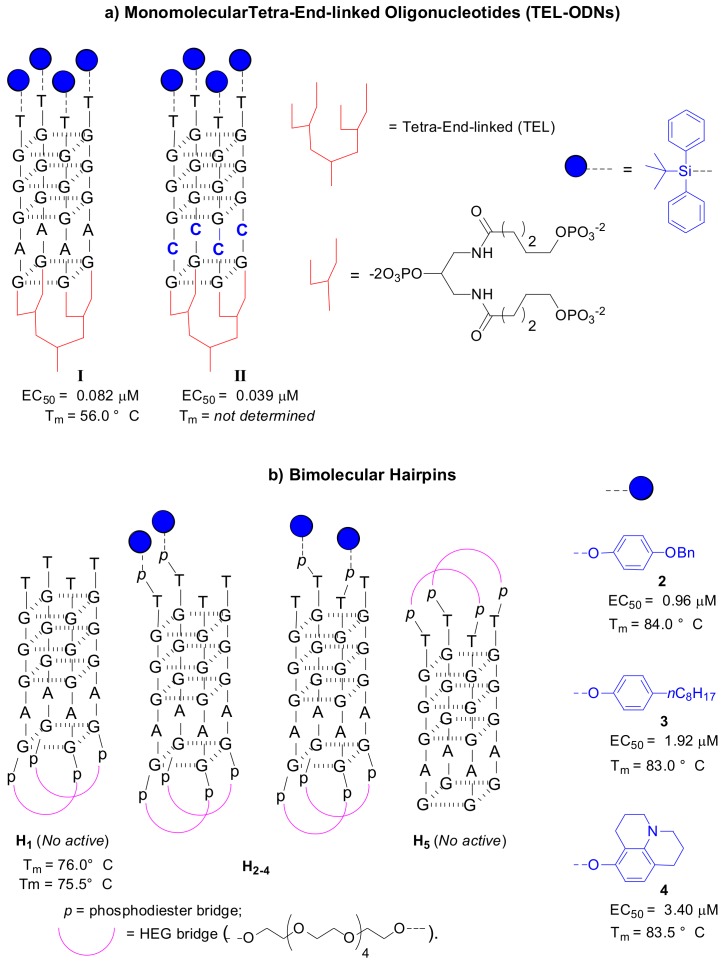
(**a**) G-quadruplex structures of monomolecular TEL-ODNs modified at 5′-end with TBDPS group (**I**) and the both 5′-end and internal modified TEL-G4 (**II**) with a cytosine that replaces the adenosine base of the Hotoda’s sequence [41,42], (**b**) Bimolecular G-quadruplex Hairpins with related anti-HIV activity (EC_50_) and thermal stability (T_m_) [43].

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
