# Peer review of "Hotoda’s Sequence and Anti-HIV Activity: Where Are We Now?"

_molecules, 2019, doi:10.3390/molecules24071417_

Round 1

Reviewer 1 Report

The mini-review entitled “The history of a strong anti-HIV agent: the Hotoda’s sequence” presented the history of a potent anti-HIV aptamer, the Hotoda’s sequence. In this review, the authors descripted the modification of Hotoda’s sequence and the biophysical, NMR and kinetic studies on natural and modified Hotoda’s sequence. It is a topic of interest to the researchers in the related areas but the paper needs some improvement before acceptance for publication. My detailed comments are as follows:

1. What’s the advantages of Hotoda’s sequence as anti-HIV agent compared to other aptamers?

2. How does the Hotoda’s sequence inhibit HIV? Is there any reports on the function mechanism or pathway of the Hotoda’s sequence?

3. Could the author try to discuss the potential of Hotoda’s sequence as anti-HIV clinical drug, as shown by the strong anti-HIV activities of Hotoda and its modified sequences?

Author Response

Reviewer 1

The mini-review entitled “The history of a strong anti-HIV agent: the Hotoda’s sequence” presented the history of a potent anti-HIV aptamer, the Hotoda’s sequence. In this review, the authors descripted the modification of Hotoda’s sequence and the biophysical, NMR and kinetic studies on natural and modified Hotoda’s sequence. It is a topic of interest to the researchers in the related areas but the paper needs some improvement before acceptance for publication. My detailed comments are as follows:

1. What’s the advantages of Hotoda’s sequence as anti-HIV agent compared to other aptamers?

Authors Response: The feature of this sequence is that was claimed to be inactive for many years, because it was not adequately studied. Now we know that even in the natural form, this sequence is strongly active with EC50 = 14 nM. We think that this result deserves our attention considering that the Hotoda’s sequence represents today one of the most active G-quadruplex anti-HIV agents [Ref. 15,45].

2. How does the Hotoda’s sequence inhibit HIV? Is there any reports on the function mechanism or pathway of the Hotoda’s sequence?

Authors Response: In the past, all studies were focused on the modified Hotoda’s sequences because they were reported to be the only active sequences. So, there are many papers that have reported potential mechanism of action of these aptamers depending on their specific modifications [Ref. 20, 22, 23, 29, 41 and 42]. Also a very recent paper [Zeye,H. et al. Journal of Pharmaceutical Sciences, 2019, doi: 10.1016/j.xphs.2019.02.008] have reported a study on the interaction of some modified of the 5’-end DMT-conjugated Hotoda’s sequence with HIV-1 surface glycoproteins in aptamer mode. (this reference has been inserted [24], lines 95-99). Regarding the natural Hotoda’s sequence, its activity was very recently discovered [2018, Ref. 45]; in this paper the authors observed a different mechanism of action for the natural d(TGGGAG) sequence respect to some 5’-end modified d(TGGGAG) sequences, but they did not study in deep this aspect.

3. Could the author try to discuss the potential of Hotoda’s sequence as anti-HIV clinical drug, as shown by the strong anti-HIV activities of Hotoda and its modified sequences?

Authors Response: The study of Hotoda’s sequence as anti-HIV clinical drug is not yet addressed. To date, the Hotoda’s sequence respect to other anti-HIV G-quadruplexes represents one of the most active G4 with an EC50 = 14 nM. As extensively discussed in the manuscript, also some modified of this sequence showed a great anti-HIV activity. It’s clear that the potentiality as clinical drug of the natural sequence respect to the modified ones could be reduced for its minor stability to the nucleases. Considering the reviewer opinion, the authors have inserted the following sentence in the manuscript at lines 375-377: “The promise of Hotoda’s sequence as anti-HIV agent in incoming to the clinical trials is alluring, the challenges are considerable but not insuperable.”

Reviewer 2 Report

In my opinion “The history...” of something could be published only as an Editorial or a Letter to the Editor. A critical review is suitable for review manuscripts. However, the manuscript is not just a diary of the Hotoda sequence and has several elements of critique and is intervened by authors’ thoughts and reflections. It is focused on relatively highly specialized, but important subject. It emphasizes the importance of non-canonical structure of nucleic acids and four-stranded DNA in particular. It is rather well-written and scientifically sound and may be found interesting not only by Molecules readers. Therefore I suggest:

1. Change the title – at least to skip the word “History”

2. Change in the style of the manuscript – avoiding describing facts from the observer point of view and include more critique and synthesis

3. Condense the manuscript

4. Avoiding the use of term like “in 1997...”, “in these years” and so on.

5. Some fragment must be clarified, e.g. lines 67-68 in Introduction, ODN must be defined when firstly used in Abstract, line 31-32 – “telomerase” or rather “telomere”?

Author Response

Reviewer 2

In my opinion “The history...” of something could be published only as an Editorial or a Letter to the Editor. A critical review is suitable for review manuscripts. However, the manuscript is not just a diary of the Hotoda sequence and has several elements of critique and is intervened by authors’ thoughts and reflections. It is focused on relatively highly specialized, but important subject. It emphasizes the importance of non-canonical structure of nucleic acids and four-stranded DNA in particular. It is rather well-written and scientifically sound and may be found interesting not only by Molecules readers. Therefore I suggest:

1.         Change the title – at least to skip the word “History”

Authors response: Thank to the reviewer for critical suggestion, we have replaced the title with “Hotoda’s Sequence and Anti-HIV Activity: Where We Are Now?”

2.         Change in the style of the manuscript – avoiding describing facts from the observer point of view and include more critique and synthesis

Authors response: The suggestion has been considered. The authors have removed and changed different sentences in the manuscript following the author suggested direction.

3.         Condense the manuscript

Authors response: The suggestion has been considered.

4.         Avoiding the use of term like “in 1997...”, “in these years” and so on.

Authors response: The corrections have been made.

5.         Some fragment must be clarified, e.g. lines 67-68 in Introduction, ODN must be defined when firstly used in Abstract, line 31-32 – “telomerase” or rather “telomere”?

Authors response: Both corrections have been made.

Round 2

Reviewer 1 Report

The manuscript has been greatly improved after revision. And all the comments from reviewers have been addressed. So I recommended its publication in the present form.